# The Feasibility and Diagnostic Adequacy of PD-L1 Expression Analysis Using the Cytoinclusion Technique in Bladder Cancer: A Prospective Single-Center Study

**DOI:** 10.3390/jcm13144072

**Published:** 2024-07-12

**Authors:** Luca Di Gianfrancesco, Isabella Monia Montagner, Debora Tormen, Alessandro Crestani, Antonio Amodeo, Paolo Corsi, Davide De Marchi, Eugenio Miglioranza, Giuliana Lista, Francesca Simonetti, Gian Maria Busetto, Martina Maggi, Filippo Marino, Antonio Scapinello, Angelo Porreca

**Affiliations:** 1Department of Urology, Veneto Institute of Oncology (IOV)—IRCCS, Headquarter of Castelfranco Veneto, 35128 Padua, Italy; luca.digianfrancesco@iov.veneto.it (L.D.G.); antonio.amodeo@iov.veneto.it (A.A.); paolo.corsi@iov.veneto.it (P.C.); davide.demarchi@iov.veneto.it (D.D.M.); eugenio.miglioranza@iov.veneto.it (E.M.); giuliana.lista@iov.veneto.it (G.L.); francesca.simonetti@iov.veneto.it (F.S.); angelo.porreca@iov.veneto.it (A.P.); 2Anatomy and Pathological Histology Unit, Veneto Institute of Oncology IOV—IRCCS, 35128 Padua, Italy; isabellamonia.montagner@iov.veneto.it (I.M.M.); debora.tormen@iov.veneto.it (D.T.); antonio.scapinello@iov.veneto.it (A.S.); 3Department of Urology, Ospedale Santa Maria Della Misericordia di Udine, 33100 Udine, Italy; alessandro.crest@gmail.com; 4Department of Urology, University of Foggia, 71122 Foggia, Italy; gianmaria.busetto@unifg.it; 5Department of Urology, Sapienza University, 00185 Rome, Italy; martina.maggi@uniroma1.it; 6Department of Urology, Fondazione Policlinico Universitario Agostino Gemelli IRCCS, University of Sacre Heart, 00168 Rome, Italy

**Keywords:** urothelial bladder cancer, PD-L1, cytoinclusion technique, urinary biomarkers

## Abstract

**Background**: Programmed death-ligand 1 (PD-L1) expression has been recognized as a potential biomarker for various cancers, yet its diagnostic and prognostic significance in urothelial bladder cancer (BCa) requires further investigation. **Methods**: In this prospective single-center study, we aimed to assess the feasibility and diagnostic adequacy of PD-L1 expression analysis using cytoinclusion in BCa patients. We enrolled consecutive patients undergoing endoscopic transurethral resection of bladder tumor (TURBT), repeat TURBT, or robot-assisted radical cystectomy. Urinary and tissue specimens were collected from these patients for cytoinclusion and histopathological analysis to evaluate PD-L1 expression. **Results**: Out of 29 patients, PD-L1 expression was detected from cytoinclusion in 42.8% (3 out of 7), 10% (1 out of 10), and 66.8% (8 out of 12) of patients with negative/papilloma, low-grade, and high-grade tumors, respectively. Conversely, histopathological analysis identified PD-L1 expression in 57.2% (4 out of 7), 30% (3 out of 10), and 83.3% (10 out of 12) of patients with negative/papilloma, low-grade, and high-grade tumors, respectively. The diagnostic concordance between cytoinclusion and histopathology was 85.7%, 80%, and 83.3% in patients with negative/papilloma, low-grade, and high-grade tumors, respectively. **Conclusions**: Our study underscores the promise of cytoinclusion as a minimally invasive method for quantifying urinary PD-L1 percentages. This approach could serve as both a potential prognostic and diagnostic indicator, easily obtainable from urine samples. Standardizing this technique could facilitate its widespread use as a valuable tool.

## 1. Introduction

Bladder cancer (BCa) remains a major contributor to global cancer-related mortality and significantly affects patient quality of life, survival rates, and healthcare costs. In 2023, BCa ranked fourth among cancer-affected men, comprising 6% of newly diagnosed cases and contributing to 4% of cancer-related deaths [1,2]. The predominant form of BCa is urothelial carcinoma of the bladder, where roughly 75% of newly identified cases consist of non-muscle-invasive bladder cancer (NMIBC), while the remaining 25% are muscle-invasive bladder cancer (MIBC) [3].

Cystoscopy combined with urine cytology is considered the gold standard for BCa detection, demonstrating a sensitivity range between 68.3% and 100% and a specificity between 75% and 97% [4]. However, the invasive nature of cystoscopy has prompted exploration into various urinary markers as potential alternatives, but none of them have been able to fully replace cystoscopy [5]. Urinary cytology has historically been pivotal as a non-invasive diagnostic tool for BCa, providing valuable insights into abnormal cell presence and aiding in diagnosing and following high-grade tumors. However, it suffers from poor sensitivity, particularly in detecting low-grade tumors [6]. Nevertheless, urinary cytology maintains a specificity of over 90% in experienced hands. It remains a fundamental examination method in the diagnostic process [7] despite its limitations, such as user-dependent interpretation and susceptibility to factors such as ongoing urinary tract infections, low cellularity, the presence of stones, or prior intravesical treatments [8].

Researchers have recently investigated the use of biomarkers as a non-invasive and effective method for detecting and surveilling BCa. Relying on a single biomarker is unlikely to address the diverse mutations and intertumoral heterogeneity. Conversely, employing biomarker panels could serve as valuable assets in categorizing patient risk and guiding treatment decisions [9]. In this context, urinary biomarkers have garnered significant attention due to their straightforward sampling process and potential to address these challenges [10,11].

A notable biomarker in this regard is programmed death-ligand 1 (PD-L1) found on the surface of cancer cells, which is pivotal in immune system evasion. Research indicates higher PD-L1 expression in BCa tumors compared to healthy bladder tissue. Furthermore, PD-L1 expression has been associated with adverse clinicopathological features and poor prognosis in BCa patients, highlighting its significance as a prognostic marker [12]. In addition to its prognostic implications, PD-L1 has also garnered attention as a predictive biomarker for immunotherapy in BCa. Immune checkpoint inhibitors targeting the PD-L1/PD-1 axis have shown promising results in clinical trials, particularly in patients with advanced or metastatic BCa. The assessment of PD-L1 expression in tumor tissue or circulating biomarkers such as urine samples may help identify patients who are likely to benefit from immunotherapy, thus guiding treatment decisions and improving clinical outcomes [12]. The assessment of PD-L1 expression in BCa has primarily relied on invasive histological specimens obtained through biopsy or endoscopic resection of bladder tumor (TURBT) [13]. However, these methods cause significant patient discomfort and require anesthesia for the procedure.

Urinary cytology offers a non-invasive option to investigate the expression of checkpoint inhibitors on exfoliated urinary cells, yet its role in this regard has not been directly assessed, and evidence in this area is limited compared to other cancers [14]. Some authors have evaluated PD-L1 expression in urinary cytology samples of patients affected by other cancers. For instance, in the study by Ya Chen et al., it was demonstrated that immunohistochemistry on urine cell blocks (UCBs) is reliable for determining PD-L1 expression in patients with upper tract urothelial carcinoma (UTUC) [15]. Additionally, the reliability of PD-L1 expression in UCBs has been shown even in studies on solid tumors, providing important prognostic and predictive information [16,17,18]. Given the promising results for these tumors, it is important to continue studying PD-L1 expression in patients with BCa.

Indeed, advancements in PD-L1 analysis on exfoliated urinary cells hold promise in enhancing our understanding of BCa’s molecular characteristics, providing prognostic information about its natural history, and potentially influencing the development of targeted intravesical therapies with checkpoint-inhibitor-based drugs. Despite these advancements, several challenges remain in the use of PD-L1 as a biomarker in BCa. The standardization of PD-L1 testing protocols, including scoring criteria and assay techniques, is crucial to ensure consistency and reliability of results across different studies and clinical settings.

In this context, our study aims to evaluate the feasibility and diagnostic adequacy of PD-L1 expression analysis using cytoinclusion in patients with BCa. The cytoinclusion technique is a minimally invasive procedure for PD-L1 analysis, involving the collection of urinary specimens. Using immunohistochemistry to evaluate PD-L1 expression in tumor cells from urinary samples assembled as UCBs can overcome the limitations of traditional methods like enzyme-linked immunosorbent assay (ELISA) analysis toll, due to the complexity of urine composition.

By exploring the correlation between PD-L1 expression in cytoinclusion and histopathological characteristics, we aim to elucidate the potential of cytoinclusion as a non-invasive diagnostic and prognostic tool for BCa.

## 2. Materials and Methods

### 2.1. Patient Population and Study Design

This study was conducted at our Italian center and received approval from the local Institutional Review Board (L02P05/2022). In total, 35 consecutive patients evaluated for BCa from July 2022 to September 2022 were screened for inclusion in this observational, prospective, and single-center study. After obtaining written informed consent, patients diagnosed with urothelial bladder cancer who underwent endoscopic transurethral resection of bladder tumor (TURBT), repeat TURBT (re-TURBT), or robot-assisted radical cystectomy (RARC) were enrolled. Exclusion criteria were ongoing chemotherapy or immunotherapy treatment for other malignancies.

Washing urinary and tissue specimens from TURBT, re-TURBT, and RARC for cytoinclusion and histopathology analysis to evaluate PD-L1 expression were collected. Specifically, the samples for cytoinclusion were obtained through urine collection at the time of TURB, re-TURB, and RARC, prior to the procedure.

PD-L1 expression was assessed using PD-L1 (SP263) rabbit monoclonal primary antibody, following the indications and instructions provided by the manufacturer. The reference standard was represented by histopathology specimens from the same patient. PD-L1 expression was evaluated based on the percentage of positive cells and staining intensity. A case was considered positive if there was expression of PD-L1 in at least one cell. Furthermore, the percentage of PD-L1 expression was calculated by dividing the number of positive cells by the total number of cells on the stained slides, following centrifugation cycles at 2500 rpm for 10 min.

Data were collected by an independent researcher not involved in the surgical procedure and histopathological analysis. The following data were collected:

Preoperative data: age, gender, body mass index (BMI), smoking habit, American Society of Anesthesiologists (ASA) score, clinical history, previous intravesical treatment, or neoadjuvant systemic treatment;

Intraoperative data: date and type of surgery, neoplasm’s focality (monofocal or multifocal), tumor size (cm), and tumor macroscopic aspect;

Postoperative data: histopathology report and tumor histotype, cytology report on urinary washing and cytoinclusion method, immunohistochemistry (IHC) method, and PD-L1 expression in histopathologic and cytology reports.

### 2.2. Statistical Analysis and Reporting

Demographic and perioperative data were analyzed using descriptive statistical techniques. Quantitative variables are presented as median and interquartile range (IQR) or mean ± standard deviation. Qualitative variables are presented as absolute and relative frequencies (percentages). We performed comparisons by Chi-square test with Yates correction or Fisher’s exact test for categorical variables and by Student’s *t*-test or Mann–Whitney’s U test, as appropriate, for continuous variables. Spearman correlation analysis and diagnostic concordance were employed to evaluate the agreement in expression levels between the two methods. Receiver operating characteristic (ROC) curve analysis and the area under the ROC curve (AUC) were used to assess the discrimination between PD-L1 expression levels using cytoinclusion and PD-L1 detection using histopathology. Statistical significance was defined as a *p*-value < 0.05. We conducted all analyses using statistical software STATA/SE version 18 (StataCorp, College Station, TX, USA).

## 3. Role of PD-L1

PD-L1, also known as B7 homolog 1 (B7-H1) or CD274, is a transmembrane protein that inhibits antitumoral T-cell responses by binding to its receptors PD-1 and B7-1 (CD80) [19].

This interaction suppresses T-cell proliferation, cytokine production, and cytolytic activity, ultimately resulting in T-cell exhaustion or inactivation [20]. Additionally, when PD-L1 binds to CD80 on T cells and antigen-presenting cells (APCs), it further downregulates immune responses by inhibiting T-cell activation and cytokine production. PD-L1 expression on tumor cells inhibits antitumor immunity, promoting immune evasion [21]. Figure 1 illustrates the mechanism of interaction between PD-L1 and CD80 in T cells. Disrupting the PD-L1/PD-1 pathway shows promise in reinvigorating tumor-specific T-cell immunity suppressed by PD-L1 expression in the tumor microenvironment. PD-L1 expression has been studied as a predictive biomarker for patients with non-small lung cancer undergoing anti-PD-1 immune checkpoint inhibitor therapy. Various cancers, including lung, melanoma, urothelial, ovarian, and colorectal cancers, exhibit PD-L1 expression with varying prevalence rates [22].

PD-L1 is commonly found in both tumor and tumor-infiltrating immune cells in patients with BCa, particularly in locally advanced cases [23]. Increased baseline tumor PD-L1 expression has been linked with poorer responses to Bacillus Calmette–Guerin (BCG) treatment in patients with NMIBC [24]. Therapeutic antibodies targeting PD-L1 or PD-1 have shown effectiveness in patients with locally advanced or metastatic BCa, resulting in the approval of five PD-1/PD-L1 inhibitors for this patient population [25].

However, the immunohistochemical assessment of PD-L1 status presents challenges due to various approved antibodies, scoring algorithms, and intratumoral heterogeneity [26]. Evaluating PD-L1 expression through IHC on histology and cytology specimens is crucial for treatment decisions involving anti-PD-1 immune checkpoint inhibitor therapy. Furthermore, the presence of PD-L1 in urinary BCa cells holds promise as a valuable diagnostic and prognostic tool. It can potentially indicate and predict the likelihood of BCa recurrence or progression, particularly when detected in exfoliated urinary cells.

## 4. Cytoinclusion Technique

The cell block technique, also known as cytoinclusion, involves converting sediments, blood clots, or visible tissue fragments from cytological samples into paraffin blocks. These blocks can then undergo cutting and staining procedures identical to those used in histopathology [27].

Cell blocks are cytologic specimens that are embedded in paraffin as blocks, similar to the method used for formalin-fixed paraffin-embedded tissue in surgical pathology. Adequate material quantity (at least 5 mL) with good cellularity is required. Unlike tissue processing in surgical pathology, the preparation of cell blocks involves various protocols that are currently in use, leading to greater variability [28].

The method used in our laboratory is described as follows: The urinary washing is fixed in formalin for 24 h and centrifuged at 2500 rpm for 10 min to remove the supernatant. After removing the supernatant, it is necessary to add 10% buffered formalin in the same proportion as the alcohol. The sample is supplemented with 70% ethanol until paraffin embedding. This step precipitates the cells present in the fluid and forms a clot. The physician can perform this step in the clinic. Upon arrival at the laboratory, the necessary processing, including dehydration and subsequent embedding in paraffin, is carried out, similar to a routine histological specimen. The sample is then cut at a thickness of 2–3 microns using a microtome, and the first and the last are stained with hematoxylin–eosin to assess the adequacy and cellularity of the samples. Subsequently, IHC staining for PD-L1 is performed in the blank sections of adequate samples to evaluate the PD-L1 expression. Human tonsil tissue is used as a positive control. Positive and negative controls are included in each staining run to ensure the accuracy and reliability of results. The stained slides are evaluated by experienced pathologists blinded to clinical data. They assess PD-L1 expression in tumor cells by calculating the proportion of tumor cells that exhibit membrane staining, regardless of the intensity (number of PD-L1 positive tumor cells/total number of tumor cells). The threshold of minimum number of neoplastic cells from cytoinclusion to be considered for the feasibility of the PD-L1 expression analysis is 20 cells.

The cytoinclusion technique offers several significant advantages in cytological and histological analysis. One of the main benefits is preserving cellular material through the use of alcohol and formalin, followed by embedding in paraffin. This process ensures optimal preservation of cells, allowing for clear and detailed visualization of cellular structures, including the nucleus and cytoplasm [28].

Furthermore, all samples are processed, mounted on a slide, and coverslipped, facilitating subsequent manipulation and analysis. This feature is particularly advantageous for performing a wide range of special stains, including immunohistochemistry, which allows for the identification of specific proteins or antigens within cells [28].

Another critical point is that cells treated with the cytoinclusion technique tend to exhibit minimal artifacts from smearing or poor preservation, such as cytoplasmic degeneration and vacuolization. As a result, both the nucleus and cytoplasm are always well visualized and well stained, providing reliable and accurate results [29].

Additionally, this technique preserves intercellular relationships, enabling the observation of any neoplastic aggregates and providing important information for tissue diagnosis and evaluation.

Finally, the cytoinclusion technique also reduces the risk of low cellularity, as all material is examined, and if necessary, subsequent trimming of the block can be performed. This means that potential areas of interest are not missed, and all relevant information can be analyzed and used for accurate diagnosis [27].

In conclusion, the cytoinclusion technique offers numerous advantages that make it a valuable choice for cytological and histological analysis. It ensures the optimal preservation of cellular material, clear visualization of cellular structures, and the ability to perform a wide range of specialized analyses.

## 5. Results

Overall, 35 patients were screened, 6 of whom were excluded due to hypocellularity after cytoinclusion analysis. A total of 29 patients were enrolled, and their data were analyzed. Table 1 shows the relevant baseline patient characteristics. TURBT, re-TURBT, and RARC were performed in 15 (51.7%), 6 (20.7%), and 8 (27.6%) patients, respectively. Cytological and histopathological analyses were performed for all washing urinary and tissue specimens; the results are illustrated in Table 2.

PD-L1 expression analysis was conducted in both washing urinary and tissue specimens. The results are summarized in Table 3. The expression of PD-L1 was identified from cytoinclusion in 42.8% (3 out of 7), 10% (1 out of 10), and 66.6% (8 out of 12) of negative/papilloma, low-grade, and high-grade patients, respectively. Conversely, in histopathological analysis, PD-L1 expression was identified in 57.2% (4 out of 7), 30% (3 out of 10), and 83.3% (10 out 12) of negative/papilloma, low-grade and high-grade patients, respectively. Considering the entire population, PD-L1 expression was identified in 41.4% (12 out of 29 patients) from cytoinclusion and 58.6% (17 out of 29 patients) from histopathology. The median values of PD-L1 expression for negative/papilloma, low-grade, and high-grade patients were 0.7%, 6%, and 2.8% in cytoinclusion and 1.6%, 5.2%, and 3.3% in histopathology, respectively.

There was no statistically significant difference (all *p* > 0.09) observed between the results obtained from the cytoinclusion analysis and those obtained from the histopathological analysis. With reference to positive cases, the degree of cellular expression is depicted in Table 4. Specifically, a higher number of PD-L1 cellular counts was observed for histopathology both in the overall population and within the individual histology categories.

Figure 2 graphically represents the proportion of patients expressing PD-L1, divided by the percentage of PD-L1 cellular count expression into three groups: <1%, between 1% and 25%, and ≥25%. It was found that in cytoinclusion, the majority of patients expressed a PD-L1 percentage of less than 25%, whereas in histopathology, a higher proportion of patients was observed.

The correlation analysis and diagnostic concordance of PD-L1 expression between the two techniques demonstrated agreement rates of 85.7%, 80%, and 83.3% in negative/papilloma, low-grade, and high-grade patients, respectively. According to the Spearman coefficient, a strong correlation was observed between the two techniques across all tumor histotype subpopulations, with an r value of 0.74, *p* < 0.001, and AUC = 0.83. Specifically, in the high-grade tumor patient group, the techniques exhibited the highest correlation (Pearson’s r = 0.58, *p* = 0.04, AUC = 0.90). Correlation values were less robust for low-grade and negative patients. Figure 3 illustrates the ROC curves, while Table 5 details the Spearman correlation results.

## 6. Discussion

BCa represents a significant financial burden and requires substantial resources within the healthcare system. The introduction of non-invasive biomarkers could be pivotal in mitigating resource strain and decreasing the expenses linked to managing this disease in healthcare systems.

Both experimental and clinical investigations have confirmed the efficacy of immune checkpoint inhibition through anti-PD-L1 antibodies in various cancer types, including BCa [30,31,32]. Despite PD-L1 being a widely utilized biomarker for both predicting outcomes and assessing responses to anti-PD-1/PD-L1 therapy in cancer, there is ongoing debate regarding its predictive and prognostic value. The expression of PD-L1 exhibits significant variability within tumors, and among patients, and this expression can change following treatments such as chemotherapy [33]. Given the dynamic nature of PD-L1 expression, continuous monitoring through serial measurements may be necessary for disease surveillance and personalized treatment adjustments. In contrast to conventional tumor tissue biopsies, blood and urine samples provide the benefit of being easily accessible.

To date, only a few studies have focused on analyzing PD-L1 expression in the urine of BCa patients. Ma et al. [12] investigated PD-L1 protein levels in preoperative urine samples from BCa patients. They evaluated the prevalence of PD-L1 in urine, explored the feasibility of using urine as a surrogate for PD-L1 expression in tumors, and compared PD-L1 expression in postoperative pathological sections. The findings indicated a strong correlation between PD-L1 levels in urine and tumor tissue, suggesting that urine could serve as a surrogate for tissues in BCa, aiding in predicting recurrence risk in muscle-invasive BCa. These results highlight the clinical significance of urine PD-L1 as a non-invasive prognostic indicator for immunotherapy and offer translational insights for developing a prognostic model for immunotherapy in BCa.

In their study, Vikerfors et al. [11] investigated soluble PD-L1 (sPD-L1) levels in serum and urine samples from 132 BCa patients and compared them to those in 51 patients with hematuria, who served as controls. Utilizing ELISA, they detected sPD-L1 in 99.5% of serum samples and 34.4% of urine samples, with a median urinary sPD-L1 concentration of 74.2 pg/mL. Urinary sPD-L1 was more frequently detected in BCa patients than controls (*p* value = 0.07), while no significant associations were found between urinary sPD-L1 levels and pT-stage or tumor grade (low vs. high), with *p* values of 0.09 for both comparisons. However, it was found that patients with metastatic disease had higher urinary levels of sPD-L1 compared to those without metastases (*p* value = 0.05). Furthermore, there was no reported association between urinary sPD-L1 levels and overall mortality (*p* = 0.09). Consequently, their study concluded that serum sPD-L1, rather than urinary sPD-L1, could potentially serve as a biomarker in BCa cases. 

In a proof-of-concept study, Tosev et al. [34] found significantly elevated levels of PD-L1 in urine samples from non-muscle-invasive BCa and muscle-invasive BCa patients compared to healthy controls. Their study reported higher median urine PD-L1 levels in newly diagnosed and recurrent BCa patients compared to controls, with statistically significant differences. Using ROC curve analysis, the authors determined optimal cutoff values for urinary PD-L1 concentrations, with sensitivity and specificity values consistent with those of FDA-approved urinary protein biomarker tests. They suggested that PD-L1 could potentially serve as a valuable adjunct biomarker alongside multiparametric panels for monitoring and detecting bladder tumors. Overall, their study provided evidence supporting the detection of PD-L1 in urine samples of BCa patients, indicating its potential utility as a biomarker for early detection, prediction, and therapeutic monitoring of BCa.

It is noteworthy that similar studies have been conducted in patients with UTUC, such as the research by Ya Chen et al. [15]. This study investigated PD-L1 expression in samples from urinary BCa and surgical resections, revealing an overall agreement of 94.4% (51 out of 54 patients). Moreover, the authors identified a PD-L1 expression cutoff (10%) in UCBs that serves as a predictive marker for the efficacy of checkpoint inhibitor immunotherapy. This highlights PD-L1’s potential as a biomarker not only for early detection but also for monitoring therapeutic outcomes in BCa.

As far as we know, there is no prior published evidence on assessing the percentage expression of PD-L1 in urine in patients with BCa. In our study, we assessed the feasibility of PD-L1 expression analysis using cytoinclusion. Our results demonstrate a high concordance rate between cytoinclusion and histopathology techniques, underscoring the potential of cytoinclusion as a promising minimally invasive tool for assessing PD-L1 expression in BCa. These findings support further research into the application of cytoinclusion.

It is important to note that the median number of PD-L1 cell counts in positive patients was markedly lower using the cytoinclusion technique compared to histopathology.

While there were variations in PD-L1 expression values through other techniques, utilizing cytoinclusion offers several notable advantages over traditional invasive procedures such as biopsy or endoscopic resection. Firstly, cytoinclusion presents an opportunity for less intrusive and more patient-friendly sampling, thereby reducing discomfort and enhancing patient compliance with follow-up assessments. This aspect is particularly significant in the context of BCa, where repeated invasive procedures can place a substantial burden on patients’ quality of life.

Secondly, cytoinclusion holds the potential to streamline diagnostic workflows by providing rapid and convenient access to cellular material for PD-L1 expression analysis. The ease of sample processing and analysis inherent in the cytoinclusion technique may contribute to improved efficiency in diagnostic laboratories, allowing for the timely evaluation of PD-L1 expression status and facilitating informed clinical decision-making.

Moreover, the non-invasive nature of cytoinclusion makes it a particularly attractive option for longitudinal monitoring of PD-L1 expression dynamics during the course of BCa treatment. By enabling serial assessments of PD-L1 expression without the need for repeated invasive interventions, cytoinclusion has the potential to enhance our understanding of the molecular evolution of BCa and its response to therapeutic interventions over time.

Recognizing the multifaceted nature of tumor-immune interactions, we acknowledge that PD-L1 represents just one component among many in BCa diagnosis. While our initial investigation prioritized PD-L1 and the specific aim of assessing cytoinclusion reliability, we acknowledge the necessity for broader exploration in future studies.

Moving forward, it is imperative to expand the scope of analysis beyond PD-L1 alone. Future research endeavors should encompass a diverse array of tumor-associated markers, delving deeper into their expression levels and employing advanced analysis software for comprehensive assessment.

The current study has some limitations, including the relatively small sample size from a single institution, which may not fully represent the heterogeneity of BCa patients. Another limitation is the use of only one anti-PD-L1 antibody; it would be beneficial to conduct comparisons using multiple types of antibodies. Furthermore, the variability in PD-L1 expression values between cytoinclusion and histopathology techniques highlights the need for further standardization and optimization of cytoinclusion protocols to ensure consistent and reliable results. The findings of this study may have limited generalizability beyond the study population, necessitating validation in different clinical settings and populations.

In our study, we acknowledge that PD-L1 expression can vary due to tumor heterogeneity, which poses challenges in biomarker assessment. However, cytoinclusion shows promise in mitigating these variations by providing access to a broader sampling of exfoliated cells, potentially offering a more representative snapshot of PD-L1 expression across different areas of the tumor.

Additional research incorporating a multi-institutional approach is warranted to validate the diagnostic accuracy and prognostic value of cytoinclusion-derived PD-L1 expression and to enhance the external validity of the cytoinclusion technique for PD-L1 analysis in BCa. Longitudinal studies assessing the clinical outcomes and therapeutic responses associated with cytoinclusion-based PD-L1 expression profiling are also needed to elucidate its potential role in guiding personalized treatment strategies and improving patient outcomes in bladder cancer. Furthermore, it is essential to perform studies correlating PD-L1 expression with immunotherapy response in patients with BCa.

## 7. Conclusions

The expression of PD-L1 on BCa cells found in urine samples could serve as both a diagnostic and prognostic indicator. It is believed that detecting PD-L1 expression on exfoliate urinary cells may provide insights into potential BCa recurrence or progression. Established analysis methods allow for the quantification of PD-L1 in urinary samples as a numerical variable. Additionally, PD-L1 expression could also be expressed in percentage terms, as in the case of the cytoinclusion technique. In order to underscore the potential significance of this novel approach, we compared cytoinclusion against the established reference standard, considering PD-L1 expression in tissue specimens. While we cannot definitively assess whether PD-L1 expression in cytoinclusion serves as a prognostic tool at this time, we aimed to explore its potential role within this context. The comparison with tissue specimen analysis may facilitate a deeper evaluation, confirming the feasibility of this technique and potentially revealing correlations between tissue analysis and cytoinclusion.

In conclusion, our study highlights the potential of PD-L1 expression on urinary cells as a dual diagnostic and prognostic marker for BCa. Detecting PD-L1 via the non-invasive cytoinclusion technique presents a promising avenue for monitoring disease recurrence and progression. We have demonstrated its comparability with established methods using tissue specimens, suggesting its utility in clinical settings. Moving forward, further research should focus on standardizing the cytoinclusion technique and validating its diagnostic accuracy through larger, multicenter studies. By harnessing the strengths of cytoinclusion while addressing its current limitations, we aim to advance personalized approaches to BCa management.

## Figures and Tables

**Figure 1 jcm-13-04072-f001:**
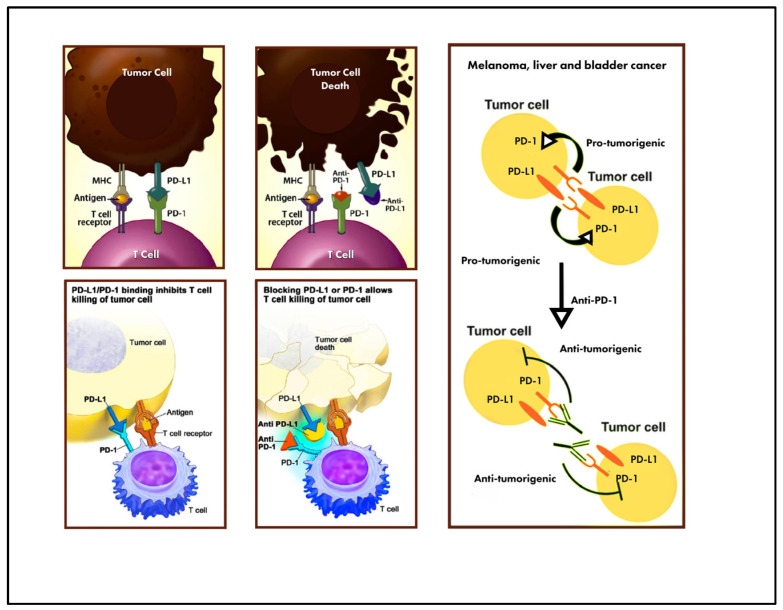
Interaction between PD-L1 and T cells. Reproduced after permission from Di Gianfrancesco et al. [10].

**Figure 2 jcm-13-04072-f002:**
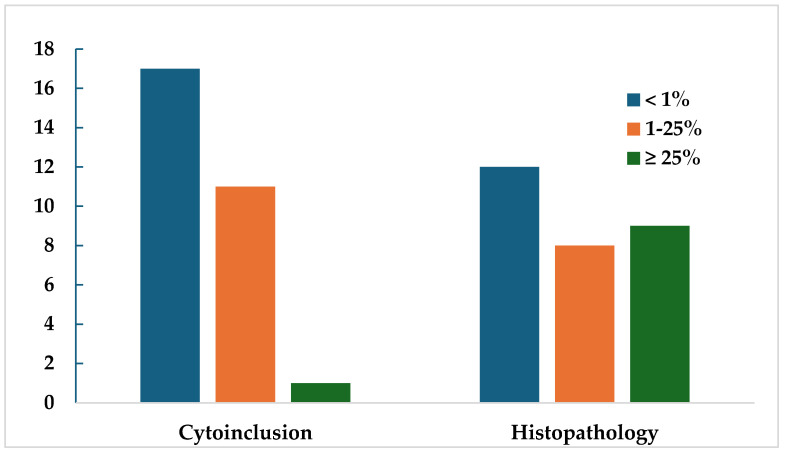
Diagram expressing the distribution of PD-L1 cellularity count using cytoinclusion and histopathology in positive PD-L1 specimens, categorized by the percentage of PD-L1 cellular count.

**Figure 3 jcm-13-04072-f003:**
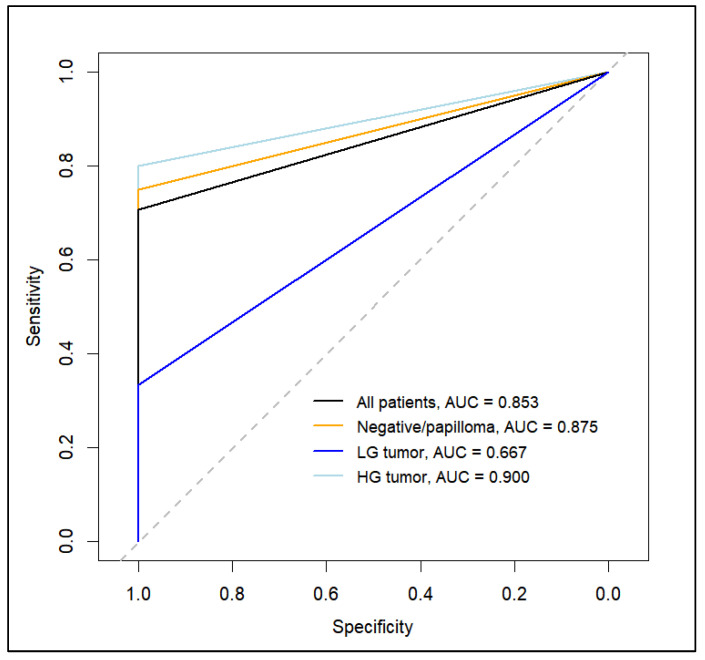
Receiving operating characteristic (ROC) curve analysis and the area under ROC curve (AUC) to assess the concordance between PD-L1 expression using cytoinclusion and histopathology for all populations and each subgroup of histology type.

**Table 1 jcm-13-04072-t001:** Baseline patient characteristics.

**Age, years**	76 (64.5–80.5)
**BMI, kg/m^2^**	28 (25–30)
**Sex, *n* (%)**	
Male	23 (79.3)
Female	6 (20.7)
**ASA score**	2 (2–2.5)
**Smoking habit, *n* (%)**	
No smoking	9 (31.1)
Former	7 (24.1)
Current smoker	13 (44.8)
**Previous intravesical treatment, *n* (%)**	
No treatment	21 (72.5)
BCG treatment	8 (27.5)
**Surgery, *n* (%)**	
re-TURBT	6 (20.7)
TURBT	15 (51.7)
RARC	8 (27.6)
**Focality, *n* (%)**	
Monofocal	17 (58.6)
Multifocal	12 (14.3)
**Tumor size, cm**	2.29 ± 1.1
**Tumor macroscopic aspect, *n* (%)**	
Papillary	21 (72.4)
Solid-papillary	5 (17.2)
Solid	3 (10.3)

BMI, body mass index; ASA, American Society of Anesthesiologists; BCG, Bacillus Calmette–Guerin; TURBT, transurethral resection of bladder tumor; RARC, robot-assisted radical cystectomy.

**Table 2 jcm-13-04072-t002:** Histopathological and cytological reports.

**Histopathology report, *n* (%)**	
Negative	5 (17.2)
Urothelial papilloma	2 (6.9)
pTaLG	10 (34.5)
pT1HG	5 (17.2)
pT2HG	4 (13.8)
pT3a/pT3b	3 (10.4)
**Histotype, *n* (%)**	
Flogosis	5 (17.2)
Pure urothelial	21(72.4)
Urothelial with squamous differentiation	3 (10.4)
**Cytology report, *n* (%)**	
NHGUC	7 (24.1)
AUC	10 (34.5)
SHGUC	3 (10.4)
HGUC	9 (31.1)

LG, low grade; HG, high grade; NHGUC, negative for high-grade urothelial carcinoma; AUC, atypical urothelial cells; SHGUC, suspicious for high-grade urothelial carcinoma; HGUC, high-grade urothelial carcinoma.

**Table 3 jcm-13-04072-t003:** PD-L1 detection using cytoinclusion and histopathology.

Histopathology Report, n (%)	PD-L1 Detection at Cytoinclusion, *n* (%)	PD-L1 Detection at Histopathology, *n* (%)	*p* Value
All patients	12 (41.4)	17 (58.6)	0.294
Negative or papilloma, 7 (24.1)	3 (42.8)	4 (57.2)	0.143
LG tumor, 10 (34.5)	1 (10)	3 (30)	0.300
HG tumor, 12 (41.4)	8 (66.6)	10 (83.3)	0.091

LG, low grade; HG, high grade; PD-L1, programmed death-ligand 1.

**Table 4 jcm-13-04072-t004:** Expression degree of PD-L1 cellularity counts using cytoinclusion and histopathology in positive PD-L1 specimens.

Histopathology Report	PD-L1 Cellular Count Using Cytoinclusion, Median (q1–q3)	PD-L1 Cellular Count Using Histopathology, Median (q1–q3)
All patients	6 (2–9)	25 (10–27)
Negative or papilloma	1 (1–1.5)	2 (1.8–2.3)
LG tumor	2	11 (10.5–11)
HG tumor	8 (7–14)	27 (25.3–27)

LG, low grade; HG, high grade; PD-L1, programmed death-ligand 1.

**Table 5 jcm-13-04072-t005:** Spearman correlation (rho) between expression degree of PD-L1 using cytoinclusion and histopathology.

	Rho	*p*-Value Rho
All patients	0.745	<0.001
Negative or papilloma	0.470	0.288
LG tumor	0.359	0.309
HG tumor	0.580	0.048

LG, low grade; HG, high grade.

## Data Availability

The data presented in this study are available upon request from the corresponding author.

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
