# Peer review of "The Feasibility and Diagnostic Adequacy of PD-L1 Expression Analysis Using the Cytoinclusion Technique in Bladder Cancer: A Prospective Single-Center Study"

_jcm, 2024, doi:10.3390/jcm13144072_

Round 1

Reviewer 1 Report

Comments and Suggestions for Authors

Title: Feasibility and Diagnostic Adequacy of PD-L1 Expression Analysis by Cytoinclusion Technique in Bladder Cancer: A Prospective Single Center Study

This study presents an interesting and relevant approach for analyzing PD-L1 expression in bladder cancer (BCa), using the cytoinclusion technique. The manuscript is well organized, offering a thorough summary of objectives, methods, and findings of this investigation; however there are certain areas in which its clarity, impact, and scientific rigor could be enhanced further.

Abstract

This abstract should provide a concise summary of your study; however, additional details could help increase clarity and completeness. A possible suggestion: Provide exact number of patients included (n=29).

- For better clarity in the conclusion, stress the potential ramifications of cytoinclusion on clinical practice.

Introduction

The Introduction sets an effective foundation, yet could be improved further by emphasizing how unique the Cytoinclusion technique is within existing literature.

Suggestion: 

- To effectively demonstrate how cytoinclusion addresses these limitations of current PD-L1 testing methods, discuss their limitations further while explaining how it overcomes them.

- Mention any previous studies which attempted to use urinary biomarkers for PD-L1 detection, providing more context for your study.

Methods

Although the methods section is detailed, some procedural aspects need further explanation to ensure reproducibility.

To do so effectively and reproducibly: Provide more specifics about centrifugation protocols - such as speeds and times.

- *Suggession: Provide more detail on the criteria used to select an adequate cytoinclusion sample for PD-L1 analysis.

Results

Went well; but for an improved interpretation and greater comprehension more context and analysis are needed to fully interpret them. It would also be useful to provide more details of statistical methods used when comparing cytoinclusion results with histopathology results.

- Suggession: To help make sense of statistical uncertainty, add confidence intervals to any percentages reported.

Discussion

These remarks are comprehensive; however, the discussion could be enhanced by exploring potential limitations and wider ramifications more thoroughly. To do so effectively: Discuss how small sample size may impede generalizability.

- Suggestion: Discuss any variations in PD-L1 expression caused by tumor heterogeneity, as well as how cytoinclusion might either mitigate or exacerbate this issue.

- Explore how cytoinclusion could fit into existing clinical workflows and compare its advantages over existing methods of care.

Figures and Tables

Figures and tables are present but could be enhanced.

One suggestion could be including a graphic abstract or flowchart summarizing the study design and key findings.

- Suggestion: Create more descriptive captions for tables and figures to ensure they can be understood without consulting the main text.

Comments Regarding Percent Match: 33%

In view of the reported percentage match, it is critical that originality and proper citation of sources are maintained. To help with this task, review your manuscript for any overlap with previously published work as well as ensure all sources are appropriately referenced.

- Suggession: When paraphrasing sections that closely resemble existing literature, focus on emphasizing your unique contributions rather than directly copying what has already been written.

Conclusion

The conclusion succinctly summarises the study while more strongly emphasizing practical implications. For future consideration, suggest emphasizing the power of cytoinclusion as an noninvasive diagnostic tool and offer specific avenues of research that build on your findings.

Overall, this manuscript presents an innovative new technique for PD-L1 analysis in BCa. By following the advice given above and making necessary modifications to their manuscript, authors can enhance its clarity, impact and scientific contribution. Further validation should occur through larger, multicenter studies in order to confirm findings and facilitate clinical translation.

Comments on the Quality of English Language

The overall quality of the English language in the manuscript is good, with clear and coherent writing.

Author Response

We thank very much the reviewer for his valuable and in-depth revision that allowed us to improve our paper. Thereafter we reported point by point the corrections/changes that have been done in the revised version according to the reviewers’ suggestion.

Reviewer 1

  1. Abstract

This abstract should provide a concise summary of your study; however, additional details could help increase clarity and completeness. A possible suggestion: Provide exact number of patients included (n=29). - For better clarity in the conclusion, stress the potential ramifications of cytoinclusion on clinical practice.

Response. Thank you for the comment. The exact number of patients included (29 patients) is stated in the “Results” subsection of the abstract. Accordingly to the reviewer’s suggestion, we have modified the Conclusion of the abstract as follows: “Our study underscores the promise of cytoinclusion as a minimally invasive method for quantifying urinary PD-L1 percentages. This approach could serve as both a potential prognostic and diagnostic indicator, easily obtainable from urine samples. Standardizing this technique could facilitate its widespread use as a valuable tool”

  1. Introduction

The Introduction sets an effective foundation, yet could be improved further by emphasizing how unique the Cytoinclusion technique is within existing literature.

Suggestion: - To effectively demonstrate how cytoinclusion addresses these limitations of current PD-L1 testing methods, discuss their limitations further while explaining how it overcomes them.

- Mention any previous studies which attempted to use urinary biomarkers for PD-L1 detection, providing more context for your study.

Response. Following the reviewer’s advice, we have revised the introduction by adding further information on the limitations of current PD-L1 testing methods as follows: “Using immunohistochemistry to evaluate PD-L1 expression in tumor cells from urinary samples assembled as UCBs can overcome the limitations of traditional methods like enzyme-linked immunosorbent assay (ELISA) analysis toll, due to the complexity of urine composition”

Additionally, we have included some studies that analyzed the detection of PD-L1 in urine for upper tract urothelial carcinoma and others solid tumors, with a more detailed examination included in the discussion section. Specifically, we have added the following: “Some authors have evaluated PD-L1 expression in urinary cytology samples of patients affected by other cancers. For instance, in the study by Ya Chen et al., it was demonstrated that immunohistochemistry on urine cell blocks (UCBs) is reliable for determining PD-L1 expression in patients with upper tract urothelial carcinoma (UTUC) [15]. Additionally, the reliability of PD-L1 expression in UCBs has been shown even in studies on solid tumors, providing important prognostic and predictive information [16-18]. Given the promising results for these tumors, it is important to continue studying PD-L1 expression in patients with BCa.”

  1. Methods

Although the methods section is detailed, some procedural aspects need further explanation to ensure reproducibility. To do so effectively and reproducibly: Provide more specifics about centrifugation protocols - such as speeds and times.

- Suggession: Provide more detail on the criteria used to select an adequate cytoinclusion sample for PD-L1 analysis.

Response. Following the reviewer’s suggestions, we have modified the Material and Methods section to include the requested information. Specifically, we have added the following details: “A case is considered positive if there is expression of PD-L1 in at least one cell. Furthermore, the percentage of PD-L1 expression is calculated by dividing the number of positive cells by the total number of cells on the stained slides, following centrifugation cycles at 2500 rpm for 10 minutes”

  1. Results

Went well; but for an improved interpretation and greater comprehension more context and analysis are needed to fully interpret them. It would also be useful to provide more details of statistical methods used when comparing cytoinclusion results with histopathology results.

- Suggestion: To help make sense of statistical uncertainty, add confidence intervals to any percentages reported.

Response. Thank you for the comment. To improve the proposed results, we performed a correlation analysis and assessed the diagnostic concordance of the agreement in expression levels between the two methods. Additionally, we presented ROC curves analysis and the area under ROC curve (AUCI). Specifically, we have added the following on statistical analysis paragraph: “Spearman correlation analysis and diagnostic concordance were employed to evaluate the agreement in expression levels between the two methods. Receiver operating characteristic (ROC) curve analysis and the area under ROC curve (AUC) were used to assess the discrimination between PD-L1 expression levels at cytoinclusion and PD-L1 detection at histopathology”. Consequently, we have updated the results with the presentation of these data and the discussion.

  1. Discussion

These remarks are comprehensive; however, the discussion could be enhanced by exploring potential limitations and wider ramifications more thoroughly. To do so effectively: Discuss how small sample size may impede generalizability.

- Suggestion: Discuss any variations in PD-L1 expression caused by tumor heterogeneity, as well as how cytoinclusion might either mitigate or exacerbate this issue.

- Explore how cytoinclusion could fit into existing clinical workflows and compare its advantages over existing methods of care.

Response. Following the reviewer’s advice, we updated the discussion to include limitations regarding sample size and the generalizability of the results. These are the statements: “The current study has some limitations including the relatively small sample size from a single institution, which may not to fully represent the heterogeneity of BCa patients” and “The findings of the study may have limited generalizability beyond the study population, necessitating validation in different clinical settings and populations.”

Furthermore, to address variations in PD-L1 expression caused by tumor heterogeneity, we have added the following: “ In our study, we acknowledge that PD-L1 expression can vary due to tumor heterogeneity, which poses challenges in biomarker assessment. However, cytoinclusion shows promise in mitigating these variations by providing access to a broader sampling of exfoliated cells, potentially offering a more representative snapshot of PD-L1 expression across different areas of the tumor”.

  1. Figures and Tables

Figures and tables are present but could be enhanced.

One suggestion could be including a graphic abstract or flowchart summarizing the study design and key findings.

- Suggestion: Create more descriptive captions for tables and figures to ensure they can be understood without consulting the main text.

Response. Following the reviewer’s guidance, we enhanced Tables 3 and 4 by including data for the entire population. Moreover, for quicker visual impact, we added ROC curves and histograms (Figure 3) illustrating the distribution of PD-L1 cellularity count via cytoinclusion and histopathology in positive patients.

  1. Comments Regarding Percent Match: 33%

In view of the reported percentage match, it is critical that originality and proper citation of sources are maintained. To help with this task, review your manuscript for any overlap with previously published work as well as ensure all sources are appropriately referenced.

- Suggession: When paraphrasing sections that closely resemble existing literature, focus on emphasizing your unique contributions rather than directly copying what has already been written.

Response. We have addressed this by making revisions and paraphrasing where necessary. The manuscript has been carefully reviewed to ensure that any sections resembling existing literature have been appropriately rephrased to emphasize our unique contributions, while maintaining originality and ensuring proper citation of all sources.

  1. Conclusion

The conclusion succinctly summarises the study while more strongly emphasizing practical implications. For future consideration, suggest emphasizing the power of cytoinclusion as an noninvasive diagnostic tool and offer specific avenues of research that build on your findings.

Overall, this manuscript presents an innovative new technique for PD-L1 analysis in BCa. By following the advice given above and making necessary modifications to their manuscript, authors can enhance its clarity, impact and scientific contribution. Further validation should occur through larger, multicenter studies in order to confirm findings and facilitate clinical translation.

Response. Thank you for your comments and suggestions regarding the conclusion of our study. We have incorporated your feedback to further emphasize the practical implications of our research, highlighting the potential of the cytoinclusion technique as a non-invasive diagnostic tool for assessing PD-L1 expression in BCa. Specifically, we have added the following: In conclusion, our study highlights the potential of PD-L1 expression on urinary cells as a dual diagnostic and prognostic marker for BCa. Detecting PD-L1 via the non-invasive cytoinclusion technique presents a promising avenue for monitoring disease recurrence and progression. We have demonstrated its comparability with established methods using tissue specimens, suggesting its utility in clinical settings. Moving forward, further research should focus on standardizing the cytoinclusion technique and validating its diagnostic accuracy through larger, multicenter studies. By harnessing the strengths of cytoinclusion while addressing its current limitations, we aim to advance personalized approaches to BCa management”

Reviewer 2 Report

Comments and Suggestions for Authors

In this work Ginafrancesco and colleagues, aim to establish a non-invasive method based on cytology to assess PD-L1 expression in patients with BC. They isolate cells from urine collected during cytology, fix and embed them into paraffin blocks, referring to this technique as cytoinclusion. Using this method, they analyze PD-L1 expression in patient samples and assess the concordance of the findings by comparing with the results with those from paired histopathological analysis samples. Below are my concerns:

1) This has the potential to be a good work, but unfortunately, it lacks novelty. This technique has already been described in the context of urothelial carcinoma in the study entitled: “Cutoff values of PD-L1 expression in urinary cytology samples for predicting response to immune checkpoint inhibitor therapy in upper urinary tract urothelial carcinoma” (https://doi.org/10.1002/cncy.22661). Although this study focuses on upper urinary tract urothelial carcinoma, the technique they present refers to the concordance of PD-L1 expression between paired surgical resection specimens (SRSs) and urine cell blocks (UCBs).

2) The authors barely present their results. Since they have performed staining of PD-L1 they do not present the immunohistochemical staining of PD-L1 in surgical resection specimens and cytoinclusion blocks. Furthermore, there is no presentation of the concordance between paired cytoinclusion blocks and surgical resection specimens with respect to PD-L1 expression in images and ROC curves.

3) If the authors have follow-up data for these patients, why don’t they correlate PD-L1 expression with response to immunotherapy?

4) I think the authors give too much emphasis and present extensively topics that are well known or can be easily found in multiple review articles e.g. the role of PD-L1. Instead of presenting these, I would recommend better explaining the results of the study.

5) The discussion should also be revised accordingly. Once the results are better presented and explained, the authors should include literature such as the one mentioned above or similar studies to compare their findings with others.

Author Response

We thank very much the reviewer for his valuable and in-depth revision that allowed us to improve our paper. Thereafter we reported point by point the corrections/changes that have been done in the revised version according to the reviewers’ suggestion.

Reviewer 2

  1. This has the potential to be a good work, but unfortunately, it lacks novelty. This technique has already been described in the context of urothelial carcinoma in the study entitled: “Cutoff values of PD-L1 expression in urinary cytology samples for predicting response to immune checkpoint inhibitor therapy in upper urinary tract urothelial carcinoma” (https://doi.org/10.1002/cncy.22661). Although this study focuses on upper urinary tract urothelial carcinoma, the technique they present refers to the concordance of PD-L1 expression between paired surgical resection specimens (SRSs) and urine cell blocks (UCBs).

Response. We thank the reviewer for his comment and for bringing to our attention this study on evaluation of cutoff values of PD-L1 expression in urinary cytology for predicting response to immune checkpoint inhibitor therapy in upper urinary tract urothelial carcinoma. Consequently, we have included it both in the introduction and the discussion to enhance our work. Specifically, we have added the following:

“Some authors have evaluated PD-L1 expression in urinary cytology samples of patients affected by other cancers. For instance, in the study by Ya Chen et al., it was demonstrated that immunohistochemistry on urine cell blocks (UCBs) is reliable for determining PD-L1 expression in patients with upper tract urothelial carcinoma (UTUC) [15]. Additionally, the reliability of PD-L1 expression in UCBs has been shown even in studies on solid tumors, providing important prognostic and predictive information [16-18]. Given the promising results for these tumors, it is important to continue studying PD-L1 expression in patients with BCa”.

  1. The authors barely present their results. Since they have performed staining of PD-L1 they do not present the immunohistochemical staining of PD-L1 in surgical resection specimens and cytoinclusion blocks. Furthermore, there is no presentation of the concordance between paired cytoinclusion blocks and surgical resection specimens with respect to PD-L1 expression in images and ROC curves.

Response. We thank the reviewer for his comment, which allowed us to improve our work. We have calculated the concordance between cytoinclusion blocks and surgical resection specimens and constructed the ROC curve. Consequently, we have revised the statistical analysis section adding the following: “Spearman correlation analysis and diagnostic concordance were employed to evaluate the agreement in expression levels between the two methods. Receiver operating characteristic (ROC) curve analysis and the area under ROC curve (AUC) were used to assess the discrimination between PD-L1 expression levels at cytoinclusion and PD-L1 detection at histopathology”

  1. If the authors have follow-up data for these patients, why don’t they correlate PD-L1 expression with response to immunotherapy?

Response. Unfortunately, we do not have this data available. We thank the reviewer for the insightful comment, and it will be considered for a future study with a larger sample size to correlate PD-L1 expression with immunotherapy response. Consequently, we have added the following to the discussion: “Furthermore, it is essential to perform studies correlating PD-L1 expression with immunotherapy response in patients with BCa”

  1. I think the authors give too much emphasis and present extensively topics that are well known or can be easily found in multiple review articles e.g. the role of PD-L1. Instead of presenting these, I would recommend better explaining the results of the study.

Response. Following the reviewer’s suggestion, we have updated the results by incorporating the correlation analysis and the ROC curve. Specifically, we have added the following in the result section: “The correlation analysis and diagnostic concordance of PD-L1 expression between the two techniques demonstrate agreement rates of 85.7%, 80%, and 83.3% in negative/papilloma, low-grade, and high-grade patients, respectively. According to the Spearman coefficient, a strong correlation was observed between the two techniques across all tumor histotype subpopulations, with an r value of 0.74, p<0.001 and AUC=0.83. Specifically, in the high-grade tumor patient group, the techniques exhibited the highest correlation (Pearson’s r=0.58, p=0.04, AUC=0.90). Correlation values were less robust for low-grade and negative patients. Figure 4 illustrates the ROC curves, while Table 5 details the Spearman correlation results”

  1. The discussion should also be revised accordingly. Once the results are better presented and explained, the authors should include literature such as the one mentioned above or similar studies to compare their findings with others.

Response. Following the reviewer’s suggestion, the discussion has been updated by adding the following: It is noteworthy that similar studies have been conducted in patients with UTUC, such as the research by Ya Chen et al. [15]. This study investigated PD-L1 expression in samples from urinary BCa and surgical resections, revealing an overall agreement of 94.4% (51 out of 54 patients). Moreover, the authors identified a PD-L1 expression cutoff (10%) in UCBs that serves as a predictive marker for the efficacy of checkpoint inhibitor immunotherapy. This highlights PD-L1’s potential as a biomarker not only for early detection but also for monitoring therapeutic outcomes in BCa”

And “As far as we know, there is no prior published evidence on assessing the percentage expression of PD-L1 in urine in patients with Bca. In our study, we assessed the feasibility of PD-L1 expression analysis on cytoinclusion. Our results demonstrate a high concordance rate between cytoinclusion and histopathology techniques, underscoring the potential of cytoinclusion as a promising mini-invasive tool for assessing PD-L1 expression in BCa. These findings support further research into the application of cytoinclusion.It is important to note that the median number of PD-L1 cell counts in positive patients is markedly lower using the cytoinclusion technique compared to histopathology”

Round 2

Reviewer 2 Report

Comments and Suggestions for Authors

The authors have tried to address all of my comments. While there are still many limitations, these are described in the discussion.